# Impact of TMB/PD-L1 expression and pneumonitis on chemoradiation and durvalumab response in stage III NSCLC

Joao V. Alessi[1], Biagio Ricciuti[1], Xinan Wang[2], Federica Pecci[1], Alessandro Di Federico[1], Giuseppe Lamberti [1], Arielle Elkrief[3,4,5], Scott J. Rodig[6,7], Emily S. Lebow[8], Jordan E. Eicholz[3], Maria Thor[9], Andreas Rimner [8], Adam J. Schoenfeld[3], Jamie E. Chaft[3], Bruce E. Johnson[1], Daniel R. Gomez[8], Mark M. Awad[1,10] & Narek Shaverdian [8,10] ✉

Although concurrent chemoradiation (CRT) and durvalumab consolidation has become a standard treatment for stage III non-small cell lung cancer (NSCLC), clinicopathologic and genomic factors associated with its efficacy remain poorly characterized. Here, in a multi-institutional retrospective cohort study of 328 patients treated with CRT and durvalumab, we identify that very high PD-L1 tumor proportion score (TPS) expression (≥ 90%) and increased tumor mutational burden (TMB) are independently associated with prolonged disease control. Additionally, we identify the impact of pneumonitis and its timing on disease outcomes among patients who discontinue durvalumab: compared to patients who experienced early-onset pneumonitis (< 3 months) leading to durvalumab discontinuation, patients with late-onset pneumonitis had a significantly longer PFS (12.7 months vs not reached; HR 0.24 [95% CI, 0.10 to 0.58]; $P$ = 0.001) and overall survival (37.2 months vs not reached; HR 0.26 [95% CI, 0.09 to 0.79]; $P$ = 0.017). These findings suggest that opportunities exist to improve outcomes in patients with lower PD-L1 and TMB levels, and those at highest risk for pneumonitis.

Despite recent advances for patients with unresectable stage III non-small cell lung cancer (NSCLC) with the addition of durvalumab (PD-L1 inhibitor) consolidation after concurrent chemoradiation (CRT), only a fraction of patients experienced durable progression-free survival (PFS)[1,2]. Discovering clinicopathologic, genomic, and immunophenotypic factors that impact outcomes to CRT and durvalumab consolidation are needed to inform future clinical trial design and lead to personalized treatment strategies.

Studies of patients with metastatic NSCLC have demonstrated that higher incremental PD-L1 tumor proportion score (TPS) levels[3–6] are associated with favorable outcomes with recent reports suggesting those with very high PD-L1 (≥ 90%) are a unique cohort with more durable disease control[6,7]. Additionally, increasing levels of tumor-associated PD-1 + CD8 + T cells[8,9], and higher tumor mutational burden (TMB)[10–12] levels are factors associated with improved efficacy to PD-(L)1 monotherapy. However, less is known about

[1]Lowe Center for Thoracic Oncology, Dana-Farber Cancer Institute, Boston, MA, USA. [2]Department of Environmental Health, Harvard T.H. Chan School of Public Health, Harvard University, Boston, MA, USA. [3]Thoracic Oncology Service, Memorial Sloan Kettering Cancer Center, New York, NY, USA. [4]Department of Pathology, Memorial Sloan Kettering Cancer Center, New York, NY, USA. [5]Human Oncology and Pathogenesis Program, Memorial Sloan Kettering Cancer Center, few York, NY, USA. [6]ImmunoProfile, Brigham and Women's Hospital, Boston, MA, USA. [7]Department of Pathology, Brigham and Women's Hospital, Boston, MA, USA. [8]Department of Radiation Oncology, Memorial Sloan Kettering Cancer Center, New York, NY, USA. [9]Department of Medical Physics, Memorial Sloan Kettering Cancer Center, New York, NY, USA. [10]These authors jointly supervised this work: Mark M. Awad, Narek Shaverdian. ✉ e-mail: shaverdn@mskcc.org

factors that influence outcomes in patients treated with CRT and durvalumab. Moreover, as CRT is known to modulate the NSCLC tumor microenvironment[13,14] specific analyses of this patient population are needed.

Recently, limited retrospective studies have reported on potential factors that can impact the efficacy of durvalumab consolidation in patients with unresectable stage III NSCLC. While patients with tumors harboring *EGFR* mutations are less likely to derive benefit with consolidation durvalumab[15], there remains conflicting and incomplete data on whether PD-L1 expression and TMB levels impact outcomes in this patient population[16]. In addition, while some data suggest that patients who experience pneumonitis have poor survival, these analyses have been limited[17]; whether the presence of pneumonitis and its timing may influence outcomes is largely unknown. With multiple trials in progress evaluating unselected approaches at treatment-intensification with combinational immunotherapy agents, which can further increase the risk of toxicities, there remains an unmet need to identify factors for patient selection.

To inform future strategies, we present a large multi-institutional analysis to identify clinical, pathologic, and genomic features that impact local-regional control, progression-free, and overall survival outcomes in patients treated with concurrent CRT and durvalumab consolidation. Specifically, we sought to assess the impact of very high ($\geq 90\%$) and negative PD-L1 TPS levels on disease outcomes, establish TMB as a predictive biomarker in this patient population and to characterize the impact of pneumonitis and its timing on outcomes. We hypothesize that we will find associations with these parameters that can be directly applied to inform future clinical trials and interventions.

## Results

### Patient clinical and pathologic features

A total of 328 patients with locally-advanced NSCLC treated with concurrent CRT and durvalumab consolidation at the DFCI ($N = 148$) and MSK ($N = 180$) were identified; their baseline clinicopathologic and genomic characteristics are included in Table 1. The median follow-up was 28.8 months (IQR: 18.7–31.0 months). The median patient age was 69 years (range: 44–86), 52% ($N = 170$) of patients were male, 95% ($N = 313$) had a history of tobacco use, and 70% had nonsquamous histology. Most patients ($N = 207$, 63%) had stage IIIB or IIIC disease. PD-L1 tumor proportion score (TPS) was available for 84% ($N = 274$) of the cohort and was < 1% among 36% of patients ($N = 98$), 1–49% among 27% ($N = 75$), 50–89% among 22% ($N = 61$), and $\geq 90\%$ among 15% ($N = 40$). Among patients with nonsquamous tumors and known *KRAS* mutational status ($N = 175$), 24% ($N = 42$) had a *KRAS*^non-G12C^ mutation and 19% ($N = 33$) had a *KRAS*^G12C^ mutation. Patients were treated with a median of 8.0 months of durvalumab (range: 1–12 months). Durvalumab consolidation started at a median of 6 weeks from end of radiation therapy (range: 1–33 weeks). A total of 115 patients completed a full year of durvalumab. Supplementary Table 1 provides information from each academic center.

### Overall treatment outcomes

Among all patients, the median OS was not reached. The 24-month OS was 75% (95% CI: 70–80%). In total, 150 patients had a progression event at a median of 6.5 months. The median PFS was 23.2 months (IQR: 14.4–40.4 months) and 24-month PFS estimate was 49% (95% CI: 43–55%). A total of 63 patients had local-regional progression at a median of 7.3 months. The 24-month local-regional control (LRC) estimate was 75% (95% CI: 69–81%).

### Impact of clinical and pathologic features on disease outcomes

The association of disease stage, age, sex, Eastern Cooperative Oncology Group performance status (ECOG PS), smoking status, tumor histology, neutrophil-to-lymphocyte ratio (NLR) and albumin levels prior durvalumab initiation, PD-L1 expression levels, and number of days between CRT end and durvalumab initiation with PFS and OS was assessed in univariable analyses (Fig. 1). Increasing disease stage was associated with a significantly shorter mPFS (IIIA: 40.6 months vs IIIB: 19.8 months vs IIIC: 12.6 months, $P = 0.002$) and mOS (IIIA: not reached vs IIIC: not reached, $P = 0.03$) (Fig. 1, Supplementary Fig. 1a). Neither patient age ( > 70 vs $\leq 70$ years), sex, smoking status, or days between CRT end to durvalumab initiation ( > 42 vs $\leq 42$ days) associated with PFS or OS (Fig. 1). Worsening ECOG PS was associated with numerically shorter mPFS (26.8 months vs 19.8 months vs 7.5 months, $P = 0.07$), and significantly shorter mOS (not reached vs

**Table 1 | Clinicopathologic and genomic characteristics of 328 patients who received durvalumab after chemoradiotherapy**

| Clinical Characteristic | *N* = 328 |
|---|---|
| Age, median (range) | 69 (44–86) |
| **Sex** | |
| Male | 170 (51.8) |
| Female | 158 (48.2) |
| **Smoking status** | |
| Current/Former | 313 (95.4) |
| Never | 15 (4.6) |
| **Histology** | |
| Nonsquamous | 228 (69.5) |
| Squamous | 100 (30.5) |
| **Oncogene diver (NSQ)^a** | |
| KRAS^b | 75 (42.9) |
| EGFR | 3 (1.7) |
| Others | 18 (10.3) |
| None identified | 79 (45.1) |
| Not assessed | 53 |
| **PD-L1 TPS** | |
| ≥90% | 40 (14.6) |
| 50–89% | 61 (22.3) |
| 1–49% | 75 (27.4) |
| <1% | 98 (35.7) |
| Not assessed | 54 |
| **Stage (AJCC 8^TH Edition)** | |
| IIIA | 121 (37.0) |
| IIIB | 158 (48.0) |
| IIIC | 49 (15.0) |
| **Radiation dose** | |
| 54–58.4 Gy | 12 (3.7) |
| 60 Gy | 265 (80.8) |
| 62–70 Gy | 51 (15.5) |
| **Chemotherapy regimen** | |
| Carboplatin + Paclitaxel | 151 (45.9) |
| Carboplatin + Pemetrexed | 75 (22.9) |
| Cisplatin + Pemetrexed | 72 (22.0) |
| Cisplatin + Etoposide | 30 (9.2) |
| **Institution** | |
| MSKCC | 180 (54.9) |
| DFCI | 148 (45.1) |

*NSQ* nonsquamous
^a175 cases with comprehensive genomic profiling
^bKRAS allele subtypes: G12C (*N* = 33), G12V (*N* = 18), G12D (*N* = 10), G13x (*N* = 4), and other KRAS (*N* = 10). Other driver mutations: ALK, BRAF, MET, and HER2

**a**

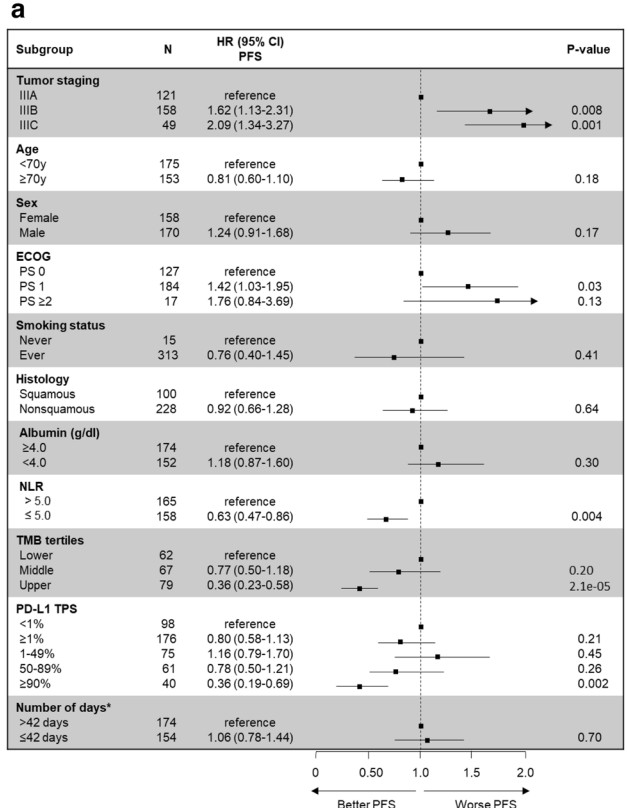

**b**

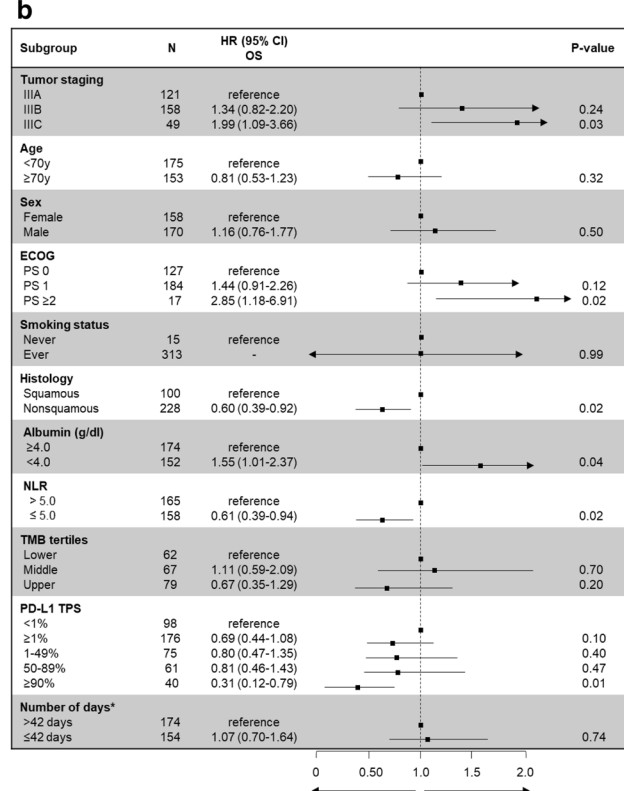

**Fig. 1 | Univariable analysis of factors associated with disease control and survival.** Forest plot for **a** progression-free and **b** overall survival with concurrent chemoradiation and durvalumab according to disease stage, age, sex, Eastern Cooperative Oncology Group performance status (ECOG PS), smoking status, tumor histology, albumin level (≥ 4.0 vs < 4.0 g/dl), neutrophil-to-lymphocyte ratio (NLR; ≥ 5.0 vs < 5.0), tumor mutation burden (TMB) tertiles, PD-L1 tumor proportion score (TPS) groups (< 1% vs 1–49% vs 50–89% vs ≥ 90%), and *number of days between radiation ends and durvalumab starts. Data are presented as the hazard ratio (HR) with error bars showing 95% confidence interval. HR and *P*-values were calculated using unadjusted Cox proportional hazard regression models. PFS progression-free survival, OS overall survival, CI confidence interval. Source data are provided as a Source Data file.

not reached vs 28.3 months, *P* = 0.04) (Fig. 1, Supplementary Fig. 1b). A higher serum albumin level (≥ 4.0 vs < 4.0 g/dl) associated with a longer OS but did not associate with PFS (Fig. 1). Among 323 cases with a complete blood count and differential prior to durvalumab initiation, the median NLR was 5.0. Patients with a higher NLR (> 5.0 vs ≤ 5.0) were found to have poor PFS and OS (16.4 vs 35.2 months, *P* = 0.004 and not reached vs not reached, *P* = 0.02, respectively) (Fig 1 and Supplementary Fig. S2).

## Impact of PD-L1 TPS on disease outcomes

On univariate analysis, among patients with PD-L1 TPS assessed (*N* = 274), PD-L1 TPS < 1% vs 1–49% and < 1% vs 50–89% did not associate with PFS or OS (*P* = 0.45, *P* = 0.26, *P* = 0.40, *P* = 0.47, respectively) (Fig. 1). However, patients with PD-L1 TPS ≥ 90% vs PD-L1 TPS < 1% had improved PFS and OS (HR 0.36, *P* = 0.002 and HR 0.31, *P* = 0.01, respectively).

Higher PD-L1 TPSs were associated with significantly longer mPFS (TPS < 1%: 14.2 months, TPS 1–49%: 12.7 months, TPS 50–89%: 24.5 months, TPS ≥ 90%: not reached, *P* = 0.002) and numerically but not significantly longer mOS (*P* = 0.08) (Fig. 2a, b). In addition, higher PD-L1 TPSs were associated with significantly improved 24-month distant control rates (50% vs 50% vs 69% vs 80%, *P* = 0.009) and numerically improved 24-month LRC (66% vs 68% vs 77% vs 92%, *P* = 0.07) (Fig. 2c). Pairwise comparisons between PD-L1 expression groups in terms of distant and local-regional outcomes are shown in Supplementary Fig. S3a, b. This analysis identified no significant difference in outcomes between patients with PD-L1 TPS < 1% and TPS 1–49% (Supplementary Fig. S3a, b).

## Efficacy of concurrent CRT and durvalumab in genomic subsets of NSCLC

We examined the impact of mutations in *TP53, KRAS, STK11, KEAP1*, and DNA-damage repair (DDR) genes on outcomes to cCRT and durvalumab, given the prevalence of these mutations and their prior associations with treatment outcomes in NSCLC[18–21]. The genomic landscape of the study population is shown in the Oncoprint (Fig. 3). First, we examined the entire study population, regardless of histology, according to *TP53* and DDR pathway mutation status. Among the 208 cases with comprehensive genomic profiling available, 139 (66.8%) had a *TP53* mutation and 155 (74.5%) had alterations identified in DDR pathway genes (Supplementary Figs S4 and S5). *TP53* mutation status associated with a significantly longer PFS, but had no impact on 24-month LRC or OS (Fig. 3). Alterations in DDR pathway genes were not found to associate with disease outcomes (Fig. 3).

Among patients with nonsquamous histology, 175 had *KRAS* mutation status available, of which 75 (42.8%) had an identified *KRAS* mutation (Supplementary Fig. S6). *KRAS* mutation status had no significant impact on 24-months LRC, PFS or OS (Fig. 3). Further analysis by *KRAS* variant subtype, found no difference in PFS or OS between *KRAS*^G12C and *KRAS*^non-G12C mutations (Supplementary Fig. S7a, b).

A total of 159 patients with nonsquamous tumors had comprehensive genomic profiling available, and were assessed for *STK11* and *KEAP1* mutation status (Supplementary Fig. S8). Patients with *STK11*^MUT tumors, compared to *STK11*^WT tumors, had a significantly shorter PFS (HR: 1.85 [95% CI, 1.16–2.96]; *P* = 0.009) but no significant impact was found on 24-month LRC or OS (Fig. 3). Patients with *KEAP1* mutation were found to have a significantly shorter OS (HR: 2.05 [95% CI,

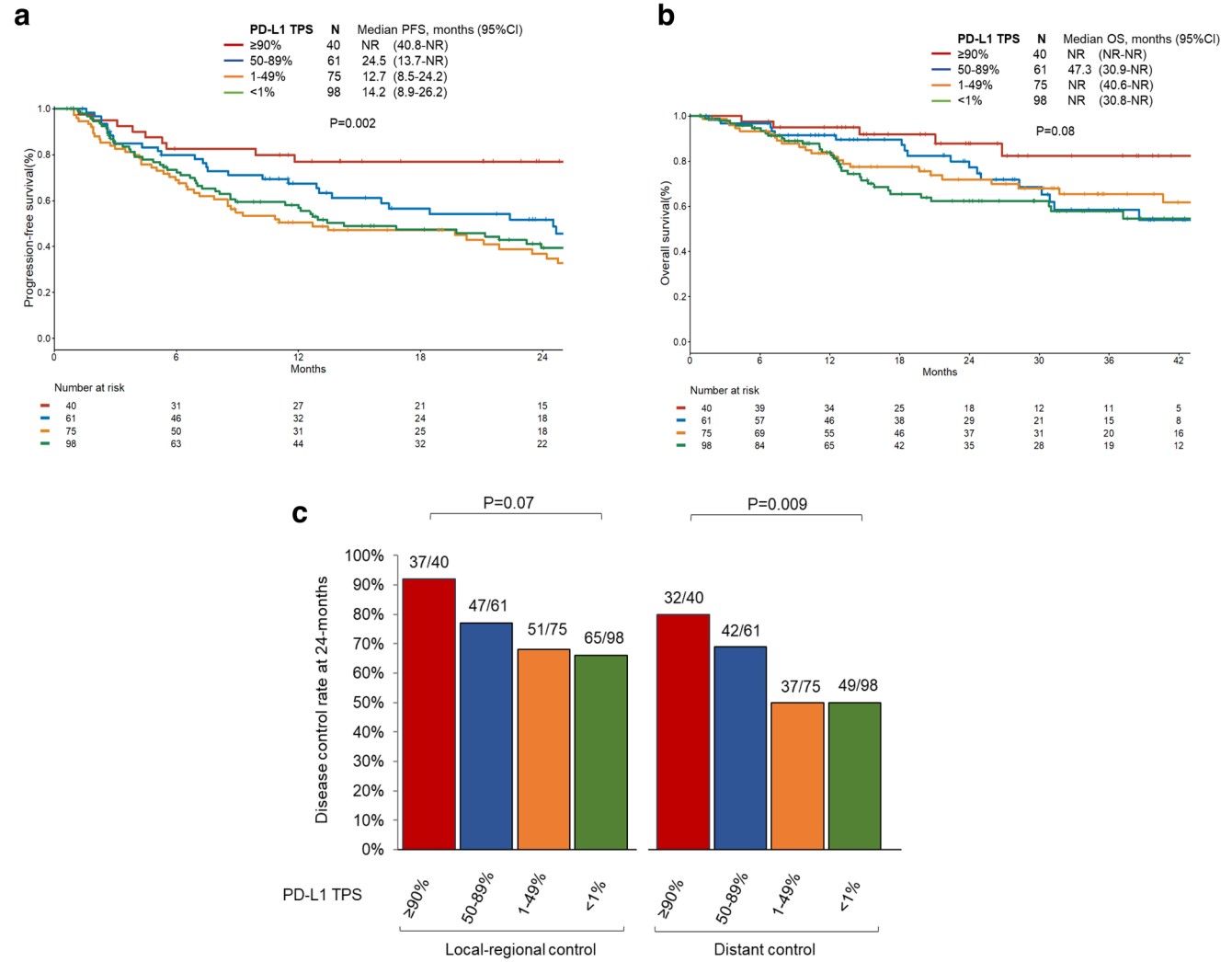

**Fig. 2 | Association between PD-L1 TPS and disease outcomes. a** Progression-free (PFS) and **b** overall survival (OS) by PD-L1 tumor proportion score (TPS) levels ( <1% vs 1−49% vs 50−89% vs ≥ 90%). **c** Local-regional and distant control rate at 24-months by PD-L1 TPS levels. *P*-values are according to log-rank test for (**a**), (**b**), and (**c**). Data are presented as median estimated control rates for (**c**). CRT chemoradiation. Source data are provided as a Source Data file.

1.02−4.12]; *P* = 0.04) but no significant difference was identified when compared to *KEAP1*[WT] in terms of 24-months LRC and PFS (Fig. 3).

As *KRAS* mutations define a subset of nonsquamous NSCLCs with heterogenous outcomes to PD-(L)1 blockade ± chemotherapy based on co-mutation status[19,22], we examined the impact of co-mutations in *TP53*, *STK11*, and *KEAP1* on PFS and OS in *KRAS*[MUT] and *KRAS*[WT] cases (Fig. 3). Among *KRAS*[WT] cases, tumors harboring *TP53*[MUT] compared with *TP53*[WT] had a significantly longer mPFS (HR 0.46; *P* = 0.01), but in *KRAS*[MUT] tumors, harboring a co-mutation in *TP53*[MUT] did not significantly impact outcomes (Fig. 3). Additionally, we observed mutations in *KEAP1* and *STK11* to have a greater negative impact on OS in patients with *KRAS*[WT] tumors (Fig. 3).

## Impact of TMB on outcomes in patients treated with concurrent CRT and durvalumab

A total of 208 patients with NSCLCs treated with concurrent CRT and durvalumab consolidation had TMB assessed [DFCI (*N* = 99) and MSK (*N* = 109)]. TMB was assayed using two different platforms (OncoPanel at DFCI and MSK-IMPACT at MSK), and therefore TMB distributions were harmonized for analysis between the two institutions by applying a normal transformation followed by standardization to Z-scores, as previously described[12,23] (Supplementary Fig. S9). On univariable analysis, increasing TMB as a continuous variable was associated with

significantly improved PFS (HR: 0.66 [95% CI, 0.55−0.78]; *P* < 0.001) but not OS (HR: 0.80 [95% CI, 0.63−1.03]; *P* = 0.08). When divided into tertiles, mPFS for lower, middle, and upper TMB tertiles were 11.0 months vs 12.6 months vs 42.0 months, *P* < 0.001 (Fig. 4a). For OS analysis, we observed no difference by TMB tertiles (47.3 months vs not reached vs not reached, *P* = 0.30, Fig. 4b). Analyses for the individual DFCI and MSKCC cohorts are shown in Supplementary Fig. S10. Of note, there was no correlation between PD-L1 TPS and TMB Z-score (Spearman *R*: 0.058, *P* = 0.46) (Supplementary Fig. S11).

Increasing TMB was also significantly associated with both localregional and distant control. When divided by TMB tertiles, 24-month local-regional control rate for lower, middle, and upper TMB tertiles was 49% vs 62% vs 85% (*P* = 0.004) and similarly, the 24-month distant control rates were 45% vs 42% vs 67% (*P* = 0.01) (Fig. 4c).

## Multivariable analysis

Multivariable Cox regression analysis confirmed that increasing disease stage and worsening ECOG PS were independently associated with shorter PFS (Fig. 5). Additionally, both PD-L1 TPS ≥ 90% and higher TMB level independently associated with increased PFS (HR: 0.43 [95% CI, 0.20−0.93], *P* = 0.032 and HR: 0.32 [95% CI, 0.19−0.54], *P* < 0.001), respectively (Fig. 5). Since genomic factors unique to nonsquamous NSCLCs were found to associated with outcomes, we performed an

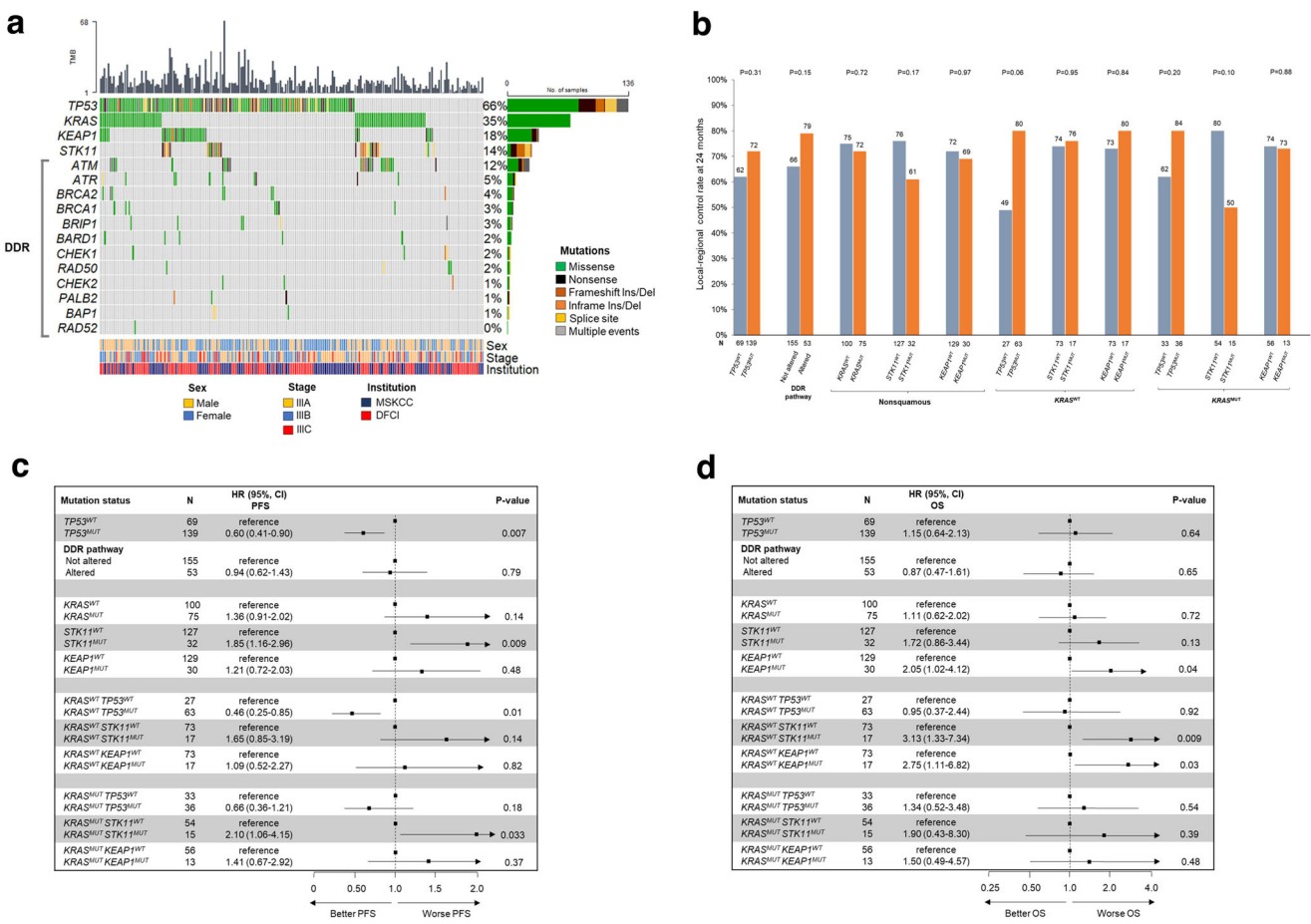

**Fig. 3 | Impact of tumor genomics on disease outcomes. a** Oncoprint of study population. **b** Local-regional control rate at 24-months, **c** progression-free, and **d** overall survival to concurrent CRT and durvalumab by *TP53* and DNA-damage repair (DDR) mutation status in NSCLC. Outcomes by *KRAS*, *STK11*, and *KEAP1* mutation status in nonsquamous NSCLC. In addition, outcomes by *TP53*, *STK11*, and *KEAP1* mutation status in *KRAS*^WT and *KRAS*^MUT nonsquamous NSCLC are also shown.

Data are presented as the hazard ratio (HR) with error bars showing 95% confidence interval. HRs were calculated with Cox proportional hazard regression for (**c**)and (**d**). *P*-values are according to log-rank test for (**b**), (**c**), and (**d**). CRT chemoradiation, HR hazard ratio, CI confidence interval, MSKCC Memorial Sloan Kettering Cancer Center, DFCI Dana-Farber Cancer Institute. Source data are provided as a Source Data file.

additional multivariable analysis in the subgroup of patients with nonsquamous histology. After adjusting for confounding factors, *TP53* and *STK11* mutations did not retain significant associations with PFS, but we found *KEAP1* mutant tumors to independently associated with poor OS (HR: 0.42 [95% CI, 0.19–0.96], *P* = 0.04) (Supplementary Fig. S12).

**Association between duration of durvalumab treatment prior to discontinuation due to pneumonitis and clinical outcomes**

Among 328 patients who received at least one dose of durvalumab, 68 (20.7%) developed treatment-related pneumonitis leading to the definitive discontinuation of durvalumab. The median time to the occurrence of pneumonitis was 77 days (range: 4–298 days). There was no difference in patient and treatment characteristics between patients who did and did not discontinue durvalumab due to pneumonitis (Supplementary Table 2). To account for lead-time bias, given the time-dependent nature of pneumonitis, we analyzed the impact of pneumonitis on survival outcomes by including pneumonitis as a time-varying co-variate in the Cox proportional hazard model and observed that the development of pneumonitis was not associated with PFS (HR: 0.95 [95% CI, 0.63–1.43], *P* = 0.79) or OS (HR: 1.14 [95% CI, 0.70–1.87], *P* = 0.60) (Supplementary Table 3).

We then disentangled the impact of pneumonitis and its timing on disease outcomes among patients who discontinued durvalumab. On multivariable analysis, a longer duration (per day unit) of durvalumab

treatment prior to discontinuation due to pneumonitis was significantly associated with increased mPFS and mOS (HR: 0.987 [95% CI, 0.979–0.995], *P* < 0.001 and HR: 0.986 [95% CI, 0.977–0.996], *P* = 0.004), respectively (Supplementary Fig. 13). Subsequently, in a time-dependent Cox regression model of patients who developed pneumonitis including its latency as an ordinal variable, the development of pneumonitis beyond 3 months after durvalumab initiation was progressively associated with a lower risk of progression and death compared to patients who developed pneumonitis within the first 3 months of durvalumab treatment (Supplementary Fig. 14a, b). In addition, visual models displaying the impact of durvalumab treatment duration on PFS and OS show an increased risk of progression/ death with shorter treatment durations followed by substantial decrease in the risk of progression/death with longer treatment durations (Supplementary Fig. 15a, b). Therefore, for subsequent analysis we used 3 months as the cutoff to define early and late-onset pneumonitis.

Compared to patients who experienced early-onset pneumonitis (*N* = 39), patients with late-onset pneumonitis (*N* = 29) had a significantly longer mPFS (12.7 months vs not reached; HR 0.24 [95% CI, 0.10 to 0.58]; *P* = 0.001; Fig. 6a) and mOS (37.2 months vs not reached; HR 0.26 [95% CI, 0.09 to 0.79]; *P* = 0.017; Fig. 6b) (Supplementary Table 4). To ensure that the duration of durvalumab treatment did not influence the differences in outcomes through lead-time bias, we examined mPFS and mOS after the development of pneumonitis for

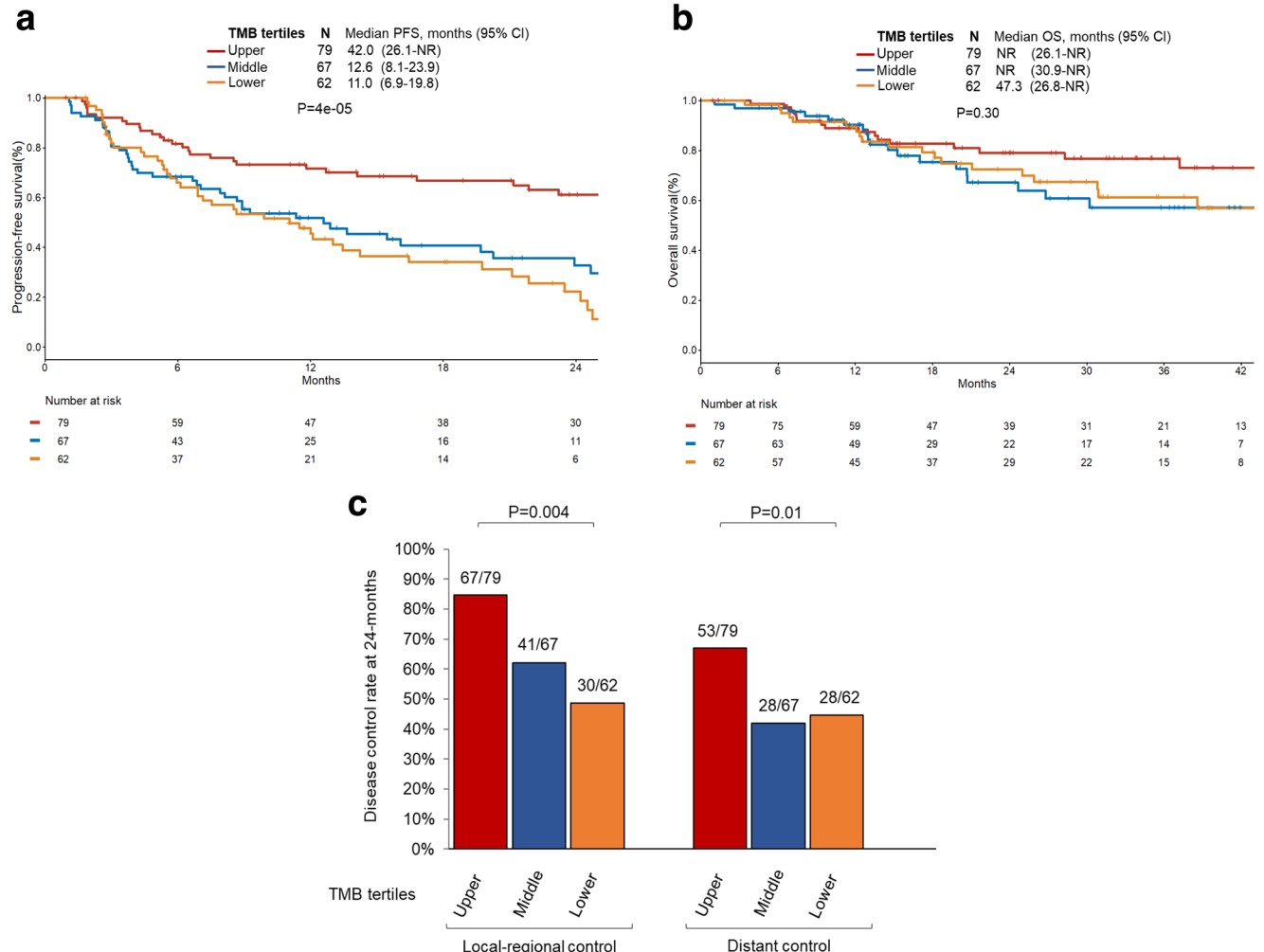

**Fig. 4 | Association between TMB and disease outcomes. a** Progression-free (PFS) and **b** overall survival (OS) by tumor mutation burden (TMB) tertiles. **c** Local-regional and distant control rate at 24-months by TMB tertiles. *P*-values are according to log-rank test for (**a**), (**b**), and (**c**). Data are presented as median estimated control rates for (**c**). CRT, chemoradiation. Source data are provided as a Source Data file.

each group. We confirmed a significantly longer mPFS and mOS among cases who experienced late-onset pneumonitis compared to early-onset pneumonitis (Supplementary Fig. 16a, b).

Having demonstrated that pneumonitis and its timing is associated with clinical outcomes to CRT and durvalumab in our multi-institutional cohort of patients, we further evaluated whether patients who experienced late-onset pneumonitis compared to 115 patients who completed a full year of durvalumab may differ in outcomes. Among 29 cases with late-onset pneumonitis, the median time to the occurrence of pneumonitis was 5.1 months (range: 3.1–9.8 months). There was no difference in mPFS and mOS between patients who experienced late-onset pneumonitis and those with full year of durvalumab treatment (Fig. 6c, d).

**Efficacy of durvalumab consolidation by baseline immunophenotypic features**

To examine whether baseline tumor-infiltrating immune cells might predict which NSCLCs are more likely to benefit from CRT and durvalumab, we interrogated immune cell subsets of NSCLCs using multiplex immunofluorescence (mIF) platform (ImmunoProfile) which quantifies CD8, FOXP3, PD-1, and PD-L1. In a subset of 21 NSCLC samples at DFCI, we observed that tumors that experienced mPFS longer than 6 months (*N* = 13) were significantly enriched in intratumoral PD-1 + CD8 + T cells (48 vs 4 cells/mm², *P* = 0.01), PD-1 + immune cells (137

vs 15 cells/mm2, *P* = 0.04), and PD-L1 positivity on immune cells (12% vs 1%, *P* = 0.04) compared to tumors that progressed earlier (*N* = 8). There were no significant differences in FOXP3 + T cells or PD-L1 positivity on tumor cells according to mPFS >6 vs ≤ 6 months (Supplementary Fig. S17a, b).

## Discussion

In this analysis of prognostic factors for patients with stage III NSCLC treated with concurrent CRT and durvalumab, our pertinent findings include that patients with very high PD-L1 expression (TPS ≥ 90%) have significantly prolonged PFS, and equally important, those with PD-L1 negative disease have outcomes similar to patients with PD-L1 low (TPS 1–49%) tumors. Additionally, we establish that TMB-high status associates with improved PFS, largely driven by improved local-regional outcomes. Finally, we also demonstrate that patients who discontinue durvalumab early due to pneumonitis have significantly worse disease control and shortened survival. These findings provide important insights to guide treatment decision making, clinical trial interpretation, and future treatment strategies.

Consistent with recent studies of PD-(L)1 monotherapy in metastatic setting, we found that very high PD-L1 expression, namely PD-L1 TPS ≥ 90%, associates with favorable treatment outcomes in patients with stage III NSCLC treated with CRT and durvalumab consolidation. While in the PACIFIC study, pre-CRT samples scored at

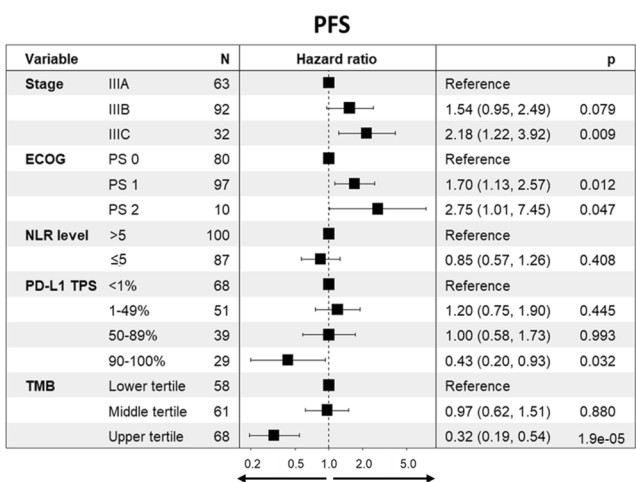

**Fig. 5 | Multivariable analysis of factors associated with disease control and survival.** Forest plot for **a** progression-free and **b** overall survival in multivariable Cox regression analysis in the cohort of patients with stage III NSCLC treated with concurrent chemoradiation and durvalumab. Data are presented as the hazard ratio (HR) with error bars showing 95% confidence interval. Cox proportional hazards regression models were applied to calculate the HR. Eastern Cooperative Oncology Group performance status (ECOG PS), neutrophil-to-lymphocyte ratio (NLR), tumor mutation burden (TMB), and PD-L1 tumor proportion score (TPS). Source data are provided as a Source Data file.

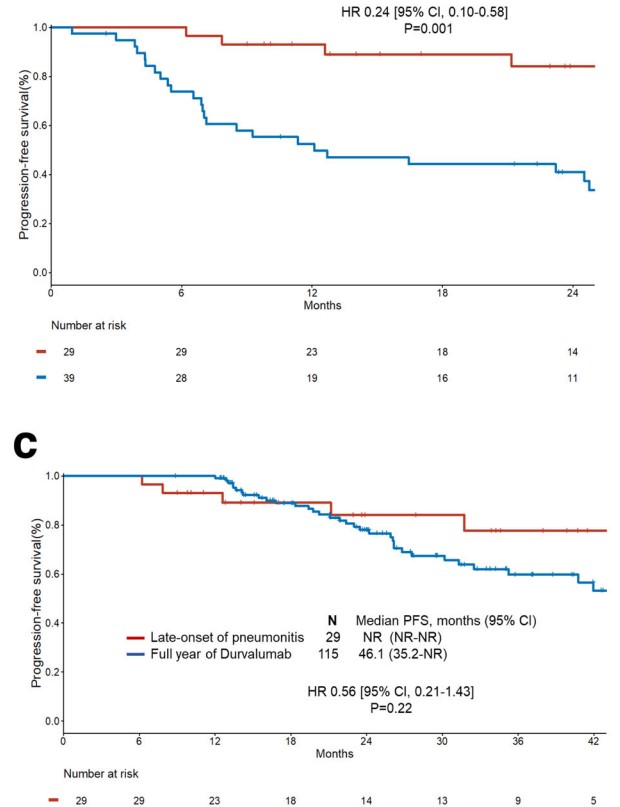

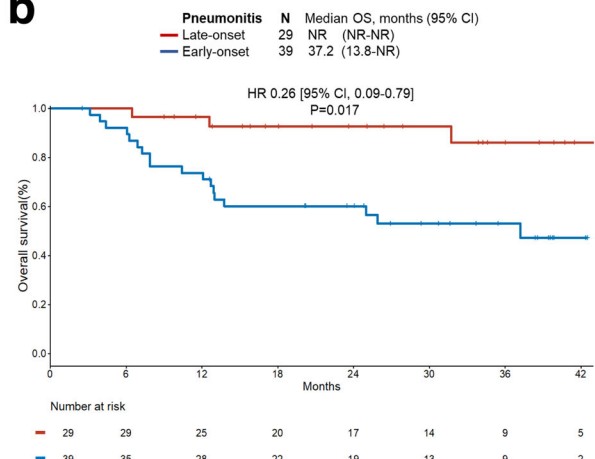

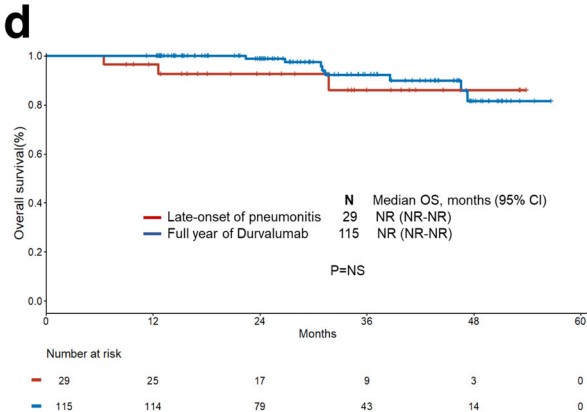

**Fig. 6 | Impact of pneumonitis on disease outcomes.** **a** Progression-free (PFS) and **b** overall survival (OS) among patients who experienced early-onset pneumonitis (< 3 months) versus late-onset pneumonitis (≥ 3 months) after durvalumab initiation. **c** PFS and **d** OS among patients who received full year of durvalumab versus patients who experienced late-onset pneumonitis. Data are presented as the hazard ratio (HR) with error bars showing 95% confidence interval (CI). HR and P-values were calculated using unadjusted Cox proportional hazard regression models for (**a**), (**b**), (**c**), and (**d**). NR not reached, NS not significant. Source data are provided as a Source Data file.

a pre-specified PD-L1 TPS 25% cutoff (< vs ≥) had similar outcomes in mPFS to durvalumab consolidation[24]. The trial report did not provide information on whether increased PD-L1 expression levels at levels as high as ≥90% also had additional benefits with durvalumab. The favorable outcomes we identified of this very high PD-L1 TPS cohort, with an unreached median PFS or OS and a 2-year local-regional control rate >90% suggests that these patients may benefit from personalized treatment strategies and that PD-L1 ≥ 90% should be introduced as a potential stratification factor into future trials if additional data confirms our initial observation. However, contrary to data from PD-(L)1 monotherapy in advanced NSCLC without preceding CRT exposure[12] we found no significant difference in outcomes between tumors with PD-L1 TPS < 1% and 1–49% in our patient cohorts. Mechanistically, CRT has been found to enhance antigen presentation and modulate the tumor microenvironment, with multiple studies finding increased density of tumor-infiltrating CD8 + T cells post CRT[25,26]. While these data may support our observation of similar clinical outcomes between patients with PD-L1 negative and PD-L1 low (1–49%) tumors and the continued use of durvalumab consolidation in this patient population, a non-inferiority study would be necessary to definitively compare outcomes between tumors with PD-L1 < 1% versus PD-L1 1–49%.

This multi-institutional data further establishes TMB as an independent predictive biomarker in stage III NSCLCs treated with CRT and durvalumab consolidation. While TMB has been found to associate with clinical outcomes to ICI in the advanced setting, data on its use in unresectable stage III patients treated with definitive multimodal therapy has thus far been limited[12]. Several lines of evidence provide rationale that TMB can be an integral biomarker in this patient population. High TMB levels have been associated with DDR mutations and increased infiltration of intratumoral CD8 + T cells[12,21,27–29]. Given that DDR genes and T cells may both play a role in radiation sensitivity[30–32], data suggest that TMB can be a unique biomarker to predict for radiation and immunotherapy sensitivity. Although in this study, the specifically assessed DDR mutations were not found to associate with improved disease control, durvalumab consolidation did appear to improve LRC in chemoradiation-resistant phenotypes such as *KEAP1* mutant tumors[33]. In addition, consistent with a recent publication of postoperative radiation therapy in surgically-resected NSCLC[34], we found significantly improved locoregional control in tumors with higher TMB levels, with an 85% vs 45% 2-year local-regional control rate in patients with high and low TMB levels, respectively. Systemic failures were also reduced in patients with TMB-high tumors. The mechanistic basis for immune response is thought to occur through a higher proportion of tumor-infiltrating immune cells, increased PD-L1 tumor expression, upregulation of innate and adaptive immune response pathways, and a distinct mutational landscape with increasing TMB[10,12]. Importantly, even when adjusting for clinical, pathological and genomic features, TMB remained significant in predicting disease control, supporting TMB as a tool to select patients for precision therapeutic approaches.

An increasing number of continuous biomarkers are associated with immunotherapy efficacy, including PD-L1 expression and TMB. Likewise, tumor-associated immune cells seem to behave in a continuous fashion in terms of therapeutic outcomes in the advanced setting. In our cohort of NSCLC samples, although only a subset of cases had their tumors profiled using multiplexed immunofluorescence, we found a significant association between increased tumor T-cell infiltration and favorable outcomes to concurrent CRT and durvalumab consolidation. Because increased levels of CD8 + T cells and PD-1 expression by CD8 + T cells within the tumor microenvironment of NSCLCs have shown improved clinical outcomes with PD-1 blockade[8], integration of TMB and PD-L1 expression with tumor-associated immune cells may refine treatment selection for patients with locally-advanced NSCLC.

This study provides critical data that early-durvalumab discontinuation due to pneumonitis is associated with poor disease control and survival. Prior studies have been limited and inconsistent on the impact of pneumonitis on clinical outcomes[17,35,36]. Here, we found that early-onset pneumonitis from durvalumab initiation negatively impacts mPFS and mOS when compared to late-onset pneumonitis (≥ 3 months). Additionally, on multivariable analysis, longer duration of durvalumab treatment prior to discontinuation due to pneumonitis was associated with improved PFS and OS. Furthermore, these findings remained significant even with examining PFS and OS from the onset pneumonitis to reduce lead-time bias. These data suggest that pneumonitis can result in insufficient durvalumab therapy which leads to poor survival; however, whether associated factors such as underlying lung disease may influence outcomes is unclear. Nonetheless, strategies to minimize and reduce the risk of pneumonitis are critical as they can potentially improve patient outcomes. Additionally, these data suggest that investigational combination immunotherapy approaches that can further increase the risk of pneumonitis need to be cautiously explored in this patient population. Clinical data suggests distinct underlying biology and prognosis between early and late immune-related adverse events in patients receiving ICIs in NSCLCs[37]. Supporting this is our finding of similar patient, disease, and treatment characteristics between patients with early-onset and late pneumonitis. However, there are major gaps in our ability to identify immunotherapy-related pneumonitis and distinguish it from radiation pneumonitis in the locally-advanced setting[38]. Nonetheless, our finding of poor disease control and survival in patients who experienced early-onset pneumonitis is critical as multiple studies continue to combine immunotherapy agents with increasing risk of pneumonitis[39].

This study of over 300 patients provides key multi-institutional data, however there are limitations. This study is limited by its retrospective nature and lack of validation from prospective clinical trials. Future studies, including either prospective validation studies or larger multi-institutional retrospective cohorts to build upon this work are necessary to further support our findings. Nonetheless, this is the largest report of clinicopathologic and genomic correlates of concurrent CRT and durvalumab efficacy in patients with stage III NSCLCs to date.

In conclusion, we analyzed clinicopathologic, genomic, and immunophenotypic features that impact outcomes for patients with stage III NSCLC treated with CRT and durvalumab consolidation and identified key independent factors with critical implications into treatment decision making and future treatment strategies. We found very high PD-L1 (≥ 90%) expression to identify a favorable patient subset, but for PD-L1 negative and PD-L1 low patient populations to have comparable outcomes. Additionally, we further established the role of TMB as an independent biomarker in this patient population and the association between high-TMB and favorable outcomes. Our data also underscore the risk of pneumonitis and the need for mitigation strategies given our findings of poor disease control and survival in patients with early-onset pneumonitis. With an emerging number of novel adjuvant therapies on the horizon, these data can inform patient selection and serve as the basis for future personalized strategies in the management of unresectable stage III NSCLC.

## Methods

### Patient population

This multicenter retrospective analysis included all consecutive patients with documented stage III NSCLC (AJCC 8th Edition) treated with platinum-based chemotherapy concurrently with definitive radiation therapy and received at least one dose of consolidation durvalumab between November 2017 and July 2022.

Clinicopathologic and genomic data were obtained from Dana-Farber Cancer Institute (DFCI) and Memorial Sloan Kettering Cancer

Center (MSKCC) cohorts. Patients were included if they had consented to each institution's institutional review board-approved medical review protocols. The patient studies were conducted according to the ethical guidelines of the Declaration of Helsinki. A total of 328 patients at DFCI ($N = 148$) and MSKCC ($N = 180$) between May 2017 to February 2022 were identified. In addition, a subset of cases also have comprehensive assessment of genomic alterations in TP53, STK11, KEAP1, and DDR pathway genes (Supplementary Figs S4–6 and S8). Lastly, a cohort of 208 patients with NSCLCs, including tumors with squamous and nonsquamous histology, had tumor mutational burden (TMB) values available at DFCI ($N = 99$) and MSKCC ($N = 109$). Data analysis was performed from November 2022 to March 2023.

## Specific site

**Dana-farber cancer institute cohort.** Patients at the Dana-Farber Cancer Institute who consented to institutional review board-approved protocols DF/HCC 02–180, 11–104, 13–364, and/or 17-000 which allowed for conducting translational research and multiplexed immunofluorescence, respectively, were also included.

## Neutrophil-to-lymphocyte ratio (NLR) and serum albumin

The most proximal complete blood count with differential and serum albumin level obtained prior to treatment initiation (up to 30 days before the first treatment with durvalumab) was extracted from electronic medical records; we retrospectively analyzed the impact of NLR (defined as the absolute neutrophil count /absolute lymphocyte count) and albumin level on clinical outcomes.

## Programmed death ligand 1 immunohistochemistry

The PD-L1 tumor proportion score (TPS) was determined by immunohistochemistry using validated anti-PD-L1 antibodies: E1L3N (Cell Signaling Technology, Danvers, MA), 22C3 (Dako North America Inc, Carpinteria, CA), and SP263 (Roche Tissue Diagnostics, Oro Valley, AZ) depending on local institutional practice. PD-L1 scores were performed on pre-treated tissue (Supplementary Table 5A).

## Multiplexed immunofluorescence (ImmunoProfile)

Multiplexed immunofluorescence (mIF) was performed on a separate cohort of NSCLCs from the DFCI to determine the immunophenotype-associated subgroups by staining 5-micron formalin-fixed, paraffin-embedded (FFPE) whole tissue sections with standard, primary antibodies sequentially and paired with a unique fluorochrome followed by staining with nuclear counterstain/4′,6-diamidino-2-phenylindole (DAPI)[40,41]. All samples were stained for PD-L1 (clone E1L3N), PD-1 (clone EPR4877[2]), CD8 (clone 4B11), and FOXP3 (clone D608R) (Supplementary Table 5B). Each sample had a single slide stained and scanned at 20× resolution by a Vectra Polaris imaging platform. Regions of Interest (ROIs) were defined for each image, and only these regions were used for quantitative image analysis. Within each ROI, InForm Image Analysis software (Perkin Elmer/ Akoya) was run to phenotype and score cells based on biomarker expression. A custom script quantified the number/ percentage of positive cells for relevant biomarkers in the intratumoral region, defined as the region of the slide consisting of tumors beyond the tumor-stroma interface. Cell count was calculated per ROI and averaged (unweighted) across ROIs, reported as count per millimeter squared ± standard error. Statistical significance of differential cell type enrichment between groups was estimated with the Wilcox Rank Sum test.

## Targeted tumor next-generation sequencing (NGS)

Non-small cell lung cancers typically underwent genomic sequencing using the following platforms at each institution: OncoPanel at DFCI[42] and MSK-IMPACT[43] at MSKCC. Additional commercial assays such as from Foundation Medicine were also included at the DFCI.

## Determination of TP53, STK11, KEAP1, and DDR pathogenic mutation status

We focused our mutational analyses on TP53, STK11, KEAP1, and the following twelve DNA-damage repair (DDR) genes most often altered in NSCLC evaluated by our NGS assays: ATM, ATR, BRCA1, BRCA2, BAP1, BARD1, BRIP1, CHEK1, CHEK2, PALB2, RAD50, and RAD52. All loss-of-function alterations in these mutations and DDR pathway genes (including nonsense, frameshift, indels, or splice site) were classified as deleterious according to OncoKB. To determine the pathogenicity of additional missense muts which were not annotated in OncoKB, we used a two-step approach. First, we retrieved all the identified missense muts in the Catalog of Somatic Mutations in Cancer (COSMIC) database. Second, we performed an in silico functional analysis using the Polymorphism Phenotyping v2 (PolyPhen-2) prediction tool to determine the functional significance of each missense mut[44]. Missense muts reported as pathogenic by COSMIC or with a PolyPhen-2 score of greater than or equal to 0.95 (probably damaging) were classified as deleterious. Patients harboring one or more deleterious DDR alteration were defined as DDR mutated.

## Tumor mutational burden assessment and harmonization

Tumor mutational burden (TMB), defined as the number of somatic, coding, base substitution, and indel mutations per megabase (Mb) of genome examined, was determined using the OncoPanel and MSK-IMPACT NGS platforms. DFCI mutation counts were divided by the number of bases covered in each OncoPanel version: v1, 0.753334 Mb; v2, 0.826167 Mb; and v3, 1.315078 Mb. For MSKCC samples, the mutation count was divided by 0.896665, 1.016478, and 1.139322 Mb for the 341-, 410-, and 468-gene panels, respectively. Because TMB was determined using two different platforms, TMB distributions were harmonized across institutions by applying a normal transformation followed by standardization to Z-scores, as previously described[23]. Power transformations were used to normalize cohort-specific TMB distributions, and Tukey's ladder of powers in the rcompanion package was used to identify the optimal transformation coefficient. Normalized distributions were then standardized into Z-scores by subtracting the transformed distribution mean and dividing by the standard deviation.

## Follow-up and treatment-related pneumonitis assessment

The clinical follow-up schedule followed standard treatment and included history, physical, and chest CT every 3–4 months for the first 2 years. Patients were defined to have treatment-related pneumonitis based on clinical and imaging findings as follows: (1) had pulmonary symptoms including dyspnea and/or cough, (2) had CT-based imaging changes involving either inside the radiated field or outside, and (c) had symptoms occur after completion of radiation and first dose of durvalumab and within the 12 cycles of immunotherapy consolidation by standard protocol. Definitive discontinuation of durvalumab was defined based on toxicity grading ≥G2 and medical oncologist ascription. Patient characteristics including age, sex, AJCC stage, smoking history, histology, and radiation treatment details were retrospectively reviewed. Patients with clinical and imaging characteristics consistent with treatment-related pneumonitis were retrospectively assessed for clinical outcomes. Toxicity grading was based on the Common Terminology Criteria for Adverse Events (CTCAE) v. 5.0 scoring system.

## Statistical analysis

Categorical and continuous variables were summarized descriptively using percentages and medians. The Wilcoxon–Rank Sum test and Kruskal–Wallis test were used to test for differences between continuous variables, and Fisher's exact test was used to test for associations between categorical variables. Event-time distributions were estimated using Kaplan–Meier methodology. Estimates 24-month

local-regional control (LRC) was calculated using package "ComparisonSurv" on R version 3.6.3[45]. Log-rank tests were used to test for differences in event-time distributions, and Cox proportional hazards models were used to estimate hazard ratios in univariable and multivariable models for PFS and OS. We first discovered the effect of baseline clinicopathological factors of tumor stage, age, sex, smoking status, histology, Eastern Cooperative Oncology Group performance status (ECOG PS), NLR, serum albumin level, TMB (lower vs middle vs upper tertile), PD-L1 TPS levels ( <1% vs 1–49% vs 50–89% vs ≥90%), and number of days between chemoradiation end and durvalumab initiation on clinical outcomes in univariable analyses. In addition, genomic alterations of interest were also evaluated. We then included the variables with $P$-value < 0.1 into the multivariable analysis to control for the potential confounding effects. All $P$-values are 2-sided and confidence intervals are at the 95% level, with significance pre-defined to be at the two-sided 0.05-level. All statistical analyses were performed using R v3.6.1 and v3.6.3.

### Reporting summary
Further information on research design is available in the Nature Portfolio Reporting Summary linked to this article.

## Data availability
All source data, including MSK-IMPACT data, are provided with this paper. Source data are provided with this paper.

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

## Acknowledgements

This research was funded in part through the NIH/NCI Cancer Center Support Grant P30 CA008748 and by Elva J. and Clayton L. McLaughlin Fund for Lung Cancer Research and V Foundation.

## Author contributions

J.V.A.: Conceptualization, Data curation, Formal analysis, Investigation, Methodology, Visualization, Roles/Writing - original draft, Writing - review & editing. B.R.: Conceptualization, Formal analysis, Methodology, Visualization, Roles/Writing - original draft, Writing - review & editing. X.W.: Methodology, Formal analysis, Writing - review & editing. F.P.: Data curation, Methodology, Validation, Writing - review & editing. A.d.F.: Data curation, Methodology, Validation, Writing - review & editing. G.L.: Data curation, Methodology, Validation, Writing - review & editing. A.E.: Roles/Writing - original draft, Writing - review & editing. S.J.R.: Methodology, Validation, Writing - review & editing. E.S.L.: Data curation, Methodology, Validation, Writing - review & editing. J.E.E.: Data curation, Validation, Writing - review & editing. M.T.: Data curation, Validation, Writing - review & editing. A.R.: Roles/Writing - original draft, Writing - review & editing. J.E.C.: Roles/Writing - original draft, Writing - review & editing. A.J.S.: Roles/Writing - original draft, Writing - review & editing. B.E.J.: Roles/Writing - original draft, Writing - review & editing. D.R.G.: Methodology, Roles/Writing - original draft, Writing - review & editing. M.M.A.: Conceptualization, Methodology, Project administration, Resources, Supervision, Validation, Writing - review & editing. N.S.: Conceptualization, Methodology, Project administration, Resources, Supervision, Validation, Writing - review & editing.

## Competing interests

N.S. reports research funding from Novartis and Varian. M.M.A serves as a consultant to Merck, Bristol-Myers Squibb, Genentech, AstraZeneca, Nektar, Maverick, Blueprint Medicine, Syndax, Abbvie, Gritstone, ArcherDX, Mirati, NextCure, and EMD Serono. Research Funding: Bristol-Myers Squibb, Lilly, Genentech, and AstraZeneca. B.E.J. receives post marketing royalties for EGFR mutation testing from Dana-Farber Cancer Institute, is a paid consultant to Novartis, Checkpoint Therapeutics, Hummingbird Diagnostics, Daichi Sankyo, Astra Zeneca, G1 Therapeutics, BlueDotBio, GSK, Hengrui Therapeutics, Simcere Pharmaceutical, and unpaid member of a steering committee for Pfizer, and receives research support from Novartis and Cannon Medical Imaging. D.R.G. has received consulting fees from Johnson and Johnson, Medtronic, AstraZeneca, and GRAIL. He has received honoraria from MedLearning Group and Varian. He has received research funding from Varian and AstraZeneca. E.S.L. has an equity interest and fiduciary role in Oncia Technologies. A.R. reports grants from Varian Medical Systems, Boehringer Ingelheim, Pfizer, Astra Zeneca, and Merck in addition to personal fees from Astra Zeneca, Merck, Cybrexa, Research to Practice, and MoreHealth, and reports non-financial support from Philips/Elekta. A.J.S reports grants from GSK, PACT pharma, Iovance Biotherapeutics, Achilles Therapeutics, Merck, and Harpoon Therapeutics and consulting fees from J&J, KSQ Therapeutics, BMS, Enara Bio, Perceptive Advisors, and Heat Biologics. J.E.C. reports grants from Merck, Brystol Myers Squibb, Genentech, and AstraZeneca. S.J.R. receives research funding from Bristol-Myers-Squibb and KITE/Gilead. S.J.R. is a member of the SAB of Immunitas Therapeutics. J.V.A., B.R., X.W., A.E., G.L., A.D.F., E.S.L., M.T., and F.P.: Nothing to disclose.
