## [Peer Review File · Nature Communications]

REVIEWER COMMENTS

Reviewer #1 (Remarks to the Author): expertise in NSCLC and immunotherapy response

Interesting and potentially clinically impactful real-world analysis of outcomes in a relatively large cohort patients from 2 institutions treated with chemoradiation therapy followed by durvalumab. The findings of the impact of high TMB and very high PD-L1 on outcomes, as well as the early vs. late pneumonitis are of interest and potential clinical utility.

A few minor comments

1. Statistical review is warranted to evaluate for multiple testing corrections
2. I would remove "we are the first to..." in the abstract
3. I would soften the conclusions that the patients with <1% PD-L1 did the "same" as 1-49 PD-L1. In the multivariate analysis, there was a monotonic improvement in the point estimate for OS in patients with increasing PD-L1 and as this was not a randomized study, no estimate of benefit from durvalumab in each category can be made. In PACIFIC, the OS for <1% PD-L1 showed no benefit, with the point estimate actually worse than control. Intermediate PD-L1 appeared better. There are wide confidence intervals, however, so firm statements about being no different can't be made without formal non-inferiority testing.
4. The data on KRAS was interesting, but begged the question about other genetic variables like STK11, KEAP1, p53, or DNA repair mutations. An exploratory analysis of these data would be very interesting.

Reviewer #2 (Remarks to the Author): clinical expertise in NSCLC

This is a well-written paper on an interesting retrospective multicenter study of the association of survival outcomes with clinical and genomic variables in stage III NSCLC patients that were treated with cCRT and durvalumab. It confirms the expected association of high PD-L1 (in the 90% or above group) with favorable outcome, also high TMB (upper third) was associated with favorable outcomes.

major items:

1. multivariable analysis: Please elaborate on variables of interest such as albumin, LD, NLR and LIPI, which have been proposed by others in the advanced stage setting.
 2. only in a minority of patients, CD8 T-cells have been investigated. Still an interesting signal was seen. The paper would much benefit from increasing the number of patients in which immunoprofiling is performed.
 3. the authors provide PD-L1, TMB, and some CD8 T-cell data, however, a description of the mechanism of action at the tumor microenvironment level with the interplay of tumor PD-L1 expression, immune cell infiltration, the presence of PD-L1 expressing immunosuppressive immune cells and TMB, supported by the study data, is lacking. It would improve the message of this study greatly if more data on this interplay would be given and elaborated on. Also, could the authors indicate how the tumor expression of PD-L1 correlated with TMB?
 4. although pneumonitis is a complication that may be triggered by durvalumab, it would more likely be triggered by radiotherapy (especially at 3 months after finishing CRT). And, it would be difficult to determine for sure whether the pneumonitis is due to radiotherapy alone, durvalumab alone or a mix. Possibly, the incidence of a durvalumab-mediated pneumonitis is higher 3 months after finishing RT, while until 3 months it may rather be due to radiotherapy alone. How did the authors determine whether pneumonitis was due to either durvalumab or radiotherapy? Could the proposed pneumonitis onset dichotomy of the study mainly be driven by this difference in mechanism of action? The appearance of immune-related AE (gr<4) was previously shown to be associated with longer PFS in advanced stage NSCLC. Could the authors elaborate on other irAE's aside from pneumonitis in their study population?
- Overall, it appears that (in the current version of the paper) the part on pneumonitis is not well connected with the previous part on PD-L1, TMB and CD8 T-cells, so the authors should attempt to

better connect and balance these 2 parts. Else, just leaving out the pneumonitis part from this paper would be sensible.

minor:

on line 184 please omit the second 'consolidation'

What are the noteworthy results?

- tumor PD-L1 expression of 90% or above is associated with favorable outcomes, as is high TMB (upper tertile). It is in line with what could be expected, but nice to have a confirmation in a large multicenter retrospective study

Will the work be of significance to the field and related fields?

- yes

How does it compare to the established literature? If the work is not original, please provide relevant references.

- original, in line with previous studies in advanced stage disease

Does the work support the conclusions and claims, or is additional evidence needed?

- yes, additional elaborations are needed on the presented data and more data is needed from the study population

Are there any flaws in the data analysis, interpretation and conclusions? Do these prohibit publication or require revision?

- no, but more data is needed

Is the methodology sound? Does the work meet the expected standards in your field?

- yes

Is there enough detail provided in the methods for the work to be reproduced?

- yes

Reviewer #3 (Remarks to the Author): statistical analysis of clinical trial response markers

Materials and Methods: There was no description of how retrospective cases were sampled. This is critical, especially as selection bias could impact interpretation. Was it all cases at the institution meeting selection criteria? Or a subset and if a subset, how were they selected?

The Results suggest that the evaluation of PD-L1 and TMB were two among many factors that were evaluated, however the strategy for evaluation should be described in Methods. This is key for understanding whether this evaluation is confirmatory of a prior hypotheses or simply highlights the two most predictive factors from among a large panel.

As the authors noted, pneumonitis co-occurs with the time-to-event outcomes and this time bias represents a flaw in interpretation of any findings that do not appropriately address this issue.

Thus, the primary pneumonitis analysis (not a secondary analysis) should be based on the landmark or time varying covariate approach. Of note, although these strategies disentangle the time bias they do not necessarily resolve the selection bias that conditions patient selection. (For instance, with landmark analysis patients have to have survived long enough to be included.)

Discussion: "while prospective validation will further support our findings" – this may not occur. The authors indicated, "These data suggest that pneumonitis can result in insufficient durvalumab therapy which leads to poor survival. Strategies to minimize and reduce the risk of pneumonitis are critical as they can potentially improve patient outcomes". The interpretation of early onset pneumonitis as causally related to poor outcomes is problematic as it may simply be a marker for the underlying mechanism that manifests in both early pneumonitis and poor prognosis.

RESPONSE TO REVIEWERS' COMMENTS

Reviewer #1

Minor comments:

1. Statistical review is warranted to evaluate for multiple testing corrections.

We greatly appreciate the reviewer's suggestion. We regret that we were not clearer in our method description. Systematic statistical review and multiple comparison corrections were performed. In addition, proportional hazards assumption was assessed using the Schoenfeld method. The following statements were added.

Page 6 (lines 16-19):

"The proportional hazards assumption was assessed with Schoenfeld residuals. All P-values are 2-sided, and confidence intervals are at the 95% level, with significance pre-defined to be at <0.05. Multiple comparison correction was performed using the Benjamini-Hochberg procedure."

2. I would remove "we are the first to..." in the abstract.

We thank the reviewer for this suggestion. We have reformulated this sentence accordingly:

Page 3, lines 6-10:

"Here, in a multi-institutional retrospective cohort study of 328 patients treated with CRT and durvalumab, we identify that very high PD-L1 tumor proportion score (TPS) expression ($\geq 90\%$) and increased tumor mutational burden (TMB) are independently associated with prolonged disease control."

3. I would soften the conclusions that the patients with <1% PD-L1 did the "same" as 1-49 PD-L1. In the multivariate analysis, there was a monotonic improvement in the point estimate for OS in patients with increasing PD-L1 and as this was not a randomized study, no estimate of benefit from durvalumab in each category can be made. In PACIFIC, the OS for <1% PD-L1 showed no benefit, with the point estimate actually worse than control. Intermediate PD-L1 appeared better. There are wide confidence intervals, however, so firm statements about being no different can't be made without formal non-inferiority testing.

We agree with the reviewer's comments regarding the limitation of retrospective analysis to assess non-inferiority, making it challenging to evaluate differences between tumors with PD-L1 <1% and 1-49%. We have toned down the suggestion that this may be possible in the manuscript.

Page 16 (lines 13-18):

"While these data may support our observation of similar clinical outcomes between patients with PD-L1 negative and PD-L1 low (1-49%) tumors and the continued use of durvalumab consolidation in this patient population, a non-inferiority study would be necessary to definitively compare outcomes between tumors with PD-L1 <1% versus PD-L1 1-49%."

4. The data on KRAS was interesting, but begged the question about other genetic variables like STK11, KEAP1, p53, or DNA repair mutations. An exploratory analysis of these data would be very interesting.

We thank the reviewer for this recommendation. We agree that this manuscript would be

strengthen by including additional genetic variables. Therefore, to be more comprehensive, we have expanded our analysis and included more genomic data using targeted next-generation sequencing (NGS) panels MSK-IMPACT and DFCI-OncoPanel. Please see below the new section entitled “**Efficacy of concurrent CRT and durvalumab in genomic subsets of NSCLC**”. In addition, we added a new **Figure 3** and **Supplementary Figures S4-S6 and S8 (Consort Diagrams)**. Moreover, the **Supplementary Methods** was edited to include how mutations were filtered out.

Pages 9 (lines 19-23), 10 (lines 1-23), and 11 (lines 1-5):

“We examined the impact of mutations in *TP53*, *KRAS*, *STK11*, *KEAP1*, and DNA-damage repair (DDR) genes on outcomes to cCRT and durvalumab, given the prevalence of these mutations and their prior associations with treatment outcomes in NSCLC^{18–21}. The genomic landscape of the study population is shown in the Oncoprint (**Figure 3**). First, we examined the entire study population, regardless of histology, according to *TP53* and DDR pathway mutation status. Among the 208 cases with comprehensive genomic profiling available, 139 (66.8%) had a *TP53* mutation and 155 (74.5%) had alterations identified in DDR pathway genes (**Supplementary Figures S4 and S5**). *TP53* mutation status associated with a significantly longer PFS, but had no impact on 24-month LRC or OS (**Figure 3**). Alterations in DDR pathway genes were not found to associate with disease outcomes (**Figure 3**).

Among patients with nonsquamous histology, 175 had *KRAS* mutation status available, of which 75 (42.8%) had an identified *KRAS* mutation (**Supplementary Figure S6**). *KRAS* mutation status had no significant impact on 24-months LRC, PFS or OS (**Figure 3**). Further analysis by *KRAS* variant subtype, found no difference in PFS or OS between *KRAS*^{G12C} and *KRAS*^{non-G12C} mutations (**Supplementary Figure S7A-B**).

A total of 159 patients with nonsquamous tumors had comprehensive genomic profiling available, and were assessed for *STK11* and *KEAP1* mutation status (**Supplementary Figure S8**). Patients with *STK11*^{MUT} tumors, compared to *STK11*^{WT} tumors, had a significantly shorter PFS (HR: 1.85 [95% CI, 1.16-2.96]; P=0.009) but no significant impact was found on 24-month LRC or OS (**Figure 3**). Patients with *KEAP1* mutation were found to have a significantly shorter OS (HR: 2.05 [95% CI, 1.02-4.12]; P=0.04) but no significant difference was identified when compared to *KEAP1*^{WT} in terms of 24-months LRC and PFS (**Figure 3**).

As *KRAS* mutations define a subset of nonsquamous NSCLCs with heterogenous outcomes to PD-(L)1 blockade +/- chemotherapy based on co-mutation status^{19,22}, we examined the impact of co-mutations in *TP53*, *STK11*, and *KEAP1* on PFS and OS in *KRAS*^{MUT} and *KRAS*^{WT} cases (**Figure 3**). Among *KRAS*^{WT} cases, tumors harboring *TP53*^{MUT} compared with *TP53*^{WT} had a significantly longer PFS (HR 0.46; P=0.01), but in *KRAS*^{MUT} tumors, harboring a co-mutation in *TP53*^{MUT} did not significantly impact outcomes (**Figure 3**). Additionally, we observed mutations in *KEAP1* and *STK11* to have a greater negative impact on OS in patients with *KRAS*^{WT} tumors (**Figure 3**).”

## Figure 3

**Figure 3. (A)** Oncoprint of study population. **(B)** Local-regional control rate at 24-months, **(C)** progression-free, and **(D)** overall survival to concurrent CRT and durvalumab by *TP53* and DNA-damage repair (DDR) mutation status in NSCLC. Outcomes by *KRAS*, *STK11*, and *KEAP1* mutation status in nonsquamous NSCLC. In addition, outcomes by *TP53*, *STK11*, and *KEAP1* mutation status in *KRAS*$^{WT}$ and *KRAS*$^{MUT}$ nonsquamous NSCLC are also shown.

Supplementary Figure 4

Supplementary Figure S4. Consort diagram showing the cohorts of patients in whom *TP53* mutation status was determined.

Supplementary Figure 5

Supplementary Figure S5. Consort diagram showing the cohorts of patients in whom DDR mutation status was determined.

Supplementary Figure 6

Supplementary Figure S6. Consort diagram showing the cohorts of patients in whom *KRAS* mutation status was determined.

Supplementary Figure 8

Supplementary Figure S8. Consort diagram showing the cohorts of patients in whom *STK11* and *KEAP1* mutation status were determined.

In addition we included in the **Supplementary Methods** a new section entitled: **Determination of TP53, STK11, KEAP1, and DDR pathogenic mutation status.**

Page 2 (lines 11-22) in the Supplementary Methods:

“We focused our mutational analyses on *TP53*, *STK11*, *KEAP1*, and the following twelve DNA-damage repair (DDR) genes most often altered in NSCLC evaluated by our NGS assays: *ATM*, *ATR*, *BRCA1*, *BRCA2*, *BAP1*, *BARD1*, *BRIP1*, *CHEK1*, *CHEK2*, *PALB2*, *RAD50*, and *RAD52*. All loss-of-function alterations in these mutations and DDR pathway genes (including nonsense, frameshift, indels, or splice site) were classified as deleterious according to OncoKB. To determine the pathogenicity of additional missense muts which were not annotated in OncoKB, we used a two-step approach. First, we retrieved all the identified missense muts in the Catalog of Somatic Mutations in Cancer (COSMIC) database. Second, we performed an in silico functional analysis using the Polymorphism Phenotyping v2 (PolyPhen-2) prediction tool to determine the functional significance of each missense mut⁵. Missense muts reported as pathogenic by COSMIC or with a PolyPhen-2 score of greater than or equal to 0.95 (probably damaging) were classified as deleterious. Patients harboring one or more deleterious DDR alteration were defined as DDR mutated.”

Because this section was incorporated into the manuscript and as described in the supplementary methods, variables with a signal ($P < 0.1$) in univariate analysis were included in the multivariable model (**Supplementary Figure S12**).

Page 12 (lines 10-16) in the Results section:

“Since genomic factors unique to nonsquamous NSCLCs were found to associated with outcomes, we performed an additional multivariable analysis in the subgroup of patients with nonsquamous histology. After adjusting for confounding factors, *TP53* and *STK11* mutations did not retain significant associations with PFS, but we found *KEAP1* mutant tumors to independently associated with poor OS (HR: 0.42 [95% CI, 0.19-0.96], $P = 0.04$) (**Supplementary Figures S12**).”

Supplementary Figure 12

Supplementary Figure S12. Forest plot for (A) progression-free and (B) overall survival in multivariable Cox regression analysis in the cohort of patients with stage III nonsquamous NSCLC treated with chemoradiation and durvalumab.

Lastly, the discussion section was also edited as follows:

Page 17 (lines 5-8) in the Results section:

“Although in this study, the specifically assessed DDR mutations were not found to associate

with improved disease control, durvalumab consolidation did appear to improve LRC in chemoradiation-resistant phenotypes such as *KEAP1* mutant cases³¹.”

Page 17 (lines 16-18) in the Results section:

“Importantly, even when adjusting for clinical, pathological and genomic features, TMB remained significant in predicting disease control, supporting TMB as a tool to select patients for precision therapeutic approaches.”

Reviewer #2

This is a well-written paper on an interesting retrospective multicenter study of the association of survival outcomes with clinical and genomic variables in stage III NSCLC patients that were treated with cCRT and durvalumab. It confirms the expected association of high PD-L1 (**in the 90% or above group**) with favorable outcome, also high TMB (upper third) was associated with favorable outcomes.

Major items:

1. Multivariable analysis: Please elaborate on variables of interest such **as albumin, LDH, NLR and LIPI**, which have been proposed by others in the advanced stage setting.

Thank you for these helpful comments and suggestions. We agree that these variables are important, and our groups have previously explored the impact of NLR in patients with advanced NSCLC.

Based upon this feedback, we have now collected and analyzed the albumin and NLR data; unfortunately, DFCI or MSKCC did not routinely measure serum lactate dehydrogenase (LDH). We have included in detail and edited in the Results section as follows:

Page 8 (lines 17-22) in the Results section:

“A higher serum albumin level (≥ 4.0 vs <4.0 g/dl) associated with a longer OS but did not associate with PFS (**Figure 1**). Among 323 cases with a complete blood count and differential prior to durvalumab initiation, the median NLR was 5.0. Patients with a higher NLR (>5.0 vs ≤ 5.0) were found to have poor PFS and OS (16.4 vs 35.2 months, $P=0.004$ and not reached vs not reached, $P=0.02$, respectively) (**Figure 1 and Supplementary Figure S2**).”

In addition, we edited the **Figure 1A-B** to include NLR and albumin levels and added a new **Supplementary Figure S2**:

Figure 1

Figure 1. Forest plot for (A) progression-free and (B) overall survival with concurrent chemoradiation and durvalumab according to disease stage, age, sex, Eastern Cooperative Oncology Group performance status (ECOG PS), smoking status, tumor histology, albumin level (≥4.0 vs <4.0 g/dl), neutrophil-to-lymphocyte ratio (NLR; ≥5.0 vs <5.0), tumor mutation burden (TMB) tertiles, PD-L1 tumor proportion score (TPS) groups (<1% vs 1-49% vs 50-89% vs ≥90%), and *number of days between radiation ends and durvalumab starts.

Supplementary Figure 2

Supplementary Figure S2. (A) Progression-free (PFS) and (B) overall survival (OS) according to neutrophil-lymphocyte ratio (NLR) prior to durvalumab initiation.

Since our multivariable analysis incorporated variables with a P-value <0.1 in univariable analysis, we included NLR and albumin level into the multivariable model accordingly. Please see below the final model (Figure 5).

Figure 5

Figure 5. Forest plot for **(A)** progression-free and **(B)** overall survival in multivariable Cox regression analysis in the cohort of patients with stage III NSCLC treated with concurrent chemoradiation and durvalumab.

2. Only in a minority of patients, CD8 T-cells have been investigated. Still an interesting signal was seen. The paper would much benefit from increasing the number of patients in which ImmunoProfile is performed.

We appreciate the reviewer’s feedback and suggestion. We have increased the cohort from 18 to 21 cases. Please see below the new results and **Supplementary Figure S17**.

Pages 14 (line 23) and 15 (lines 1-7) in the Results section:

“In a subset of 21 NSCLC samples at DFCI, we observed that tumors that experienced mPFS longer than 6 months (N=13) were significantly enriched in intratumoral PD-1+ CD8+ T cells (48 vs 4 cells/mm², P=0.01), PD-1+ immune cells (137 vs 15 cells/mm², P=0.04), and PD-L1 positivity on immune cells (12% vs 1%, P=0.04) compared to tumors that progressed earlier (N=8). There were no significant differences in FOXP3+ T cells or PD-L1 positivity on tumor cells according to mPFS >6 vs ≤6 months (**Supplementary Figure S17A-B**).“

Supplementary Figure S17

Supplementary Figure S17. (A) Median number of tumor-associated immune cells (CD8+, double positive PD-1+ CD8+, FOXP3+, and PD-1+ immune cells) and **(B)** PD-L1 expression on tumors and immune cells in NSCLCs from patients who experienced mPFS \geq or $<$ 6 months as best response to durvalumab. **(C)** Multiplexed immunofluorescence using the ImmunoProfile platform on 3 samples from NSCLCs.

3. The authors provide PD-L1, TMB, and some CD8 T-cell data, however, a description of the mechanism of action at the tumor microenvironment level with the interplay of tumor PD-L1 expression, immune cell infiltration, the presence of PD-L1 expressing immunosuppressive immune cells and TMB, supported by the study data, is lacking. It would improve the message of this study greatly if more data on this interplay would be given and elaborated on. Also, could the authors indicate how the tumor expression of PD-L1 correlated with TMB?

Thank you for your question on the correlation between PD-L1 expression and TMB. Because DFCI and MSKCC have collaborated on other projects, we leveraged data from 1548 patients with NSCLC that had both biomarkers assessed before immunotherapy-based therapies, including anti-PD-(L)1 monotherapy, chemoimmunotherapy, and CRT followed by durvalumab consolidation. In the entire cohort of patients, there was a very weak positive correlation between PD-L1 TPS and TMB Z-score (Spearman R : 0.050, $P=0.03$). Please see below.

While in the subgroup treated with CRT and durvalumab consolidation, we observed a similar correlation coefficient as in the cohort of 1548 cases, no significant correlation was found between the two biomarkers (N=191; [$R=0.058$, $P=0.46$]).

Supplementary Figure S11

Supplementary Figure 11. Spearman’s correlation coefficient between PD-L1 TPS (%) and TMB Z-score as continuous variable.

We added the following sentence and supplementary figure:

Page 11 (lines 21-22) in the Results section:

“Of note, there was no correlation between PD-L1 TPS and TMB Z-score (Spearman R : 0.058, $P=0.46$) (**Supplementary Figure S11**).”

In addition, we included a paragraph in the discussion section to elaborate on the correlation between biomarkers and outcomes to cCRT and durvalumab.

Pages 17 (lines 19-23) and 18 (lines 1-6) in the Discussion section:

“An increasing number of continuous biomarkers are associated with immunotherapy efficacy, including PD-L1 expression and TMB. Likewise, tumor-associated immune cells seem to behave in a continuous fashion in terms of therapeutic outcomes in the advanced setting. In our cohort of NSCLC samples, although only a subset of cases had their tumors profiled using multiplexed immunofluorescence, we found a significant association between increased tumor T-cell infiltration and favorable outcomes to concurrent CRT and durvalumab consolidation. Because increased levels of CD8 +T cells and PD-1 expression by CD8 +T cells within the tumor microenvironment of NSCLCs have shown improved clinical outcomes with PD-

1 blockade⁸, integration of TMB and PD-L1 expression with tumor-associated immune cells may refine treatment selection for patients with locally-advanced NSCLC.”

4. Although pneumonitis is a complication that may be triggered by durvalumab, it would be more likely to be triggered by radiotherapy (especially at 3 months after finishing CRT). And, it would be difficult to determine for sure whether the pneumonitis is due to radiotherapy alone, durvalumab alone or a mix. Possibly, the incidence of a durvalumab-mediated pneumonitis is higher 3 months after finishing RT, while until 3 months it may rather be due to radiotherapy alone. How did the authors determine whether pneumonitis was due to either durvalumab or radiotherapy? Could the proposed pneumonitis onset dichotomy of the study mainly be driven by this difference in mechanism of action? The appearance of immune-related AE (gr<4) was previously shown to be associated with longer PFS in advanced stage NSCLC. Could the authors elaborate on other irAE's aside from pneumonitis in their study population? Overall, it appears that (in the current version of the paper) the part on pneumonitis is not well connected with the previous part on PD-L1, TMB and CD8 T-cells, so the authors should attempt to better connect and balance these 2 parts. Else, just leaving out the pneumonitis part from this paper would be sensible.

Thank you for these helpful comments and suggestions with regards to pneumonitis.

We agree with Reviewer #2 about the challenge of determining the underlying cause of pneumonitis in these patients treated with multimodality therapy. Our manuscript aims to identify whether longer duration of durvalumab treatment before discontinuation may impact outcomes rather than the underlying mechanism triggering pneumonitis. Indeed, the inability to accurately disentangle radiation pneumonitis from ICI-mediated pneumonitis is shown in recent trials where pneumonitis is an endpoint and is not specified as radiation or ICI-mediated (NICOLAS Trial, PMID 33188912; KEYNOTE 799, PMID 32086039). We, however, do agree with Reviewer #2's suspicions that the incidence of durvalumab-mediated pneumonitis may be higher 3 months after finishing RT.

In this analysis, we focused on pneumonitis as in our combined dataset, approximately 21% of patients discontinued durvalumab therapy due to pneumonitis. This is a large number of patients and clinically significant. Our hope in this manuscript is to provide key variables that associate with outcomes to allow for better stratification and future trial design. Therefore, we do think it is important to keep the pneumonitis data with regards to durvalumab discontinuation in this manuscript.

We acknowledge that the determination of the underlying mechanism of pneumonitis may provide additional information, but data are limited regarding the interplay between radiation- and immunotherapy-induced lung injury, and which biomarkers might be associated with toxicity. Nonetheless, we dissected the impact of duration of immunotherapy treatment and outcomes before discontinuation, including radiation design.

In addition, we edited the following statements and included the below figure to highlight our aims and findings.

Page 13 (lines 18-22) in the Results section:

“In addition, visual models displaying the impact of durvalumab treatment duration on PFS and OS show an increased risk of progression/death with shorter treatment durations followed by substantial decreases in the risk of progression/death with longer treatment durations (Supplementary Figure 15A-B).”

Supplementary Figure 15

Supplementary Figure S15. Hazard ratio of duration of durvalumab treatment from **(A)** progression-free survival (PFS) and **(B)** overall survival (OS) in univariable Cox model. Restricted cubic spline was applied to duration of durvalumab treatment with the reference of 3 months (early vs late-onset pneumonitis). The shaded area is the 95% confidence interval from the restricted-cubic-spline model.

We have already highlighted the following points in the Results section:

Pages 18 (lines 22-23) and 19 (lines 1-7) in the Results section:

“Clinical data suggests distinct underlying biology and prognosis between early and late immune-related adverse events in patients receiving ICIs in NSCLCs³⁵. Supporting this is our finding of similar patient, disease, and treatment characteristics between patients with early-onset and late pneumonitis. However, there are major gaps in our ability to identify immunotherapy-related pneumonitis and distinguish it from radiation pneumonitis in the locally-advanced setting³⁶. Nonetheless, our finding of poor disease control and survival in patients who experienced early-onset pneumonitis is critical as multiple studies continue to combine immunotherapy agents with increasing risk of pneumonitis³⁷.”

minor:

on line 184 please omit the second 'consolidation'
Thank you for the reviewer. We corrected this typo.

Reviewer #3 (Remarks to the Author): statistical analysis of clinical trial response markers

1. Materials and Methods: There was no description of how retrospective cases were sampled. This is critical, especially as selection bias could impact interpretation. Was it all cases at the institution meeting selection criteria? Or a subset and if a subset, how were they selected? Thank you for bringing up this question. We made clearer the study population and inclusion criteria in the main manuscript and supplementary methods as follows:

Page 5 (lines 17-20) in the Material and Methods section:

“This retrospective analysis included all consecutive patients with documented stage III NSCLC (AJCC 8th Edition) treated with platinum-based chemotherapy concurrently with definitive radiation therapy and received at least one dose of durvalumab consolidation between November 2017 to July 2022.”

Page 1 (lines 5-17) in the Supplementary Methods, Patient population section:

“This multicenter retrospective analysis included all consecutive patients with documented stage III NSCLC (AJCC 8th Edition) treated with platinum-based chemotherapy concurrently with definitive radiation therapy and received at least one dose of consolidation durvalumab between November 2017 to July 2022. Clinicopathologic and genomic data were obtained from Dana-Farber Cancer Institute (DFCI) and Memorial Sloan Kettering Cancer Center (MSKCC) cohorts. Patients were included if they had consented to each institution's institutional review board-approved medical review protocols. The patient studies were conducted according to the ethical guidelines of the Declaration of Helsinki. A total of 328 patients at DFCI (N=148) and MSKCC (N=180) between November 2017 to July 2022 were identified. In addition, a subset of cases also have comprehensive assessment of genomic alterations in *TP53*, *STK11*, *KEAP1*, and DDR pathway genes (**Supplementary Figures S4-S6 and S8**). Lastly, a cohort of 208 patients with NSCLCs, including tumors with squamous and nonsquamous histology, had tumor mutational burden (TMB) values available at DFCI (N=99) and MSKCC (N=109). Data analysis was performed from November 2022 to March 2023.”

2. **The Results** suggest that the evaluation of PD-L1 and TMB were two among many factors that were evaluated, however the strategy for evaluation should be described in Methods. This is key for understanding whether this evaluation is confirmatory of a prior hypotheses or simply highlights the two most predictive factors from among a large panel.

Thank you for these questions. The Supplementary Methods describes in great detail how both biomarker were assessed, including TMB harmonization across different next-generation sequencing panels. Please see below.

Page 1 (lines 36-39) in the Supplementary Methods:

“The PD-L1 tumor proportion score (TPS) was determined by immunohistochemistry using validated anti-PD-L1 antibodies: E1L3N (Cell Signaling Technology, Danvers, MA), 22C3 (Dako North America Inc, Carpinteria, CA), and SP263 [Roche Tissue Diagnostics, Oro Valley, AZ] depending on local institutional practice. PD-L1 scores were performed on pre-treated tissue.”

Page 2 (lines 25-35) in the Supplementary Methods:

“Tumor mutational burden (TMB), defined as the number of somatic, coding, base substitution, and indel mutations per megabase (Mb) of genome examined, was determined using the OncoPanel and MSK-IMPACT NGS platforms. DFCI mutation counts were divided by the number of bases covered in each OncoPanel version: v1, 0.753334 Mb; v2, 0.826167 Mb; and v3, 1.315078 Mb. For MSKCC samples, the mutation count was divided by 0.896665, 1.016478, and 1.139322 Mb for the 341-, 410-, and 468-gene panels, respectively. Because TMB was determined using two different platforms, TMB distributions were harmonized across institutions by applying a normal transformation followed by standardization to Z-scores, as previously described⁶. Power transformations were used to normalize cohort-specific TMB distributions, and Tukey's ladder of powers in the rcompanion package was used to identify the optimal transformation coefficient. Normalized distributions were then standardized into Z-scores by subtracting the transformed distribution mean and dividing by the standard deviation. “

In addition, Supplementary Methods describes the strategy for evaluation of these variables as follows.

Page 3 (lines 8-14) in the Statistical Analysis section:

“We first discovered the effect of baseline clinicopathological factors of tumor stage, age, sex,

smoking status, histology, Eastern Cooperative Oncology Group performance status (ECOG PS), NLR, serum albumin level, TMB (lower vs middle vs upper tertile), PD-L1 TPS levels (<1% vs 1-49% vs 50-89% vs ≥90%), and number of days between chemoradiation end and durvalumab initiation on clinical outcomes in univariable analyses. In addition, genomic alterations of interest were also evaluated. We then included the variables with P-value <0.1 into the multivariable analysis to control for the potential confounding effects.”

3. As the authors noted, pneumonitis co-occurs with the time-to-event outcomes and this time bias represents a flaw in interpretation of any findings that do not appropriately address this issue. Thus, the primary pneumonitis analysis (not a secondary analysis) should be based on the landmark or time varying covariate approach. Of note, although these strategies disentangle the time bias they do not necessarily resolve the selection bias that conditions patient selection. (For instance, with landmark analysis patients have to have survived long enough to be included.)

Thank you for bringing up this question. We included analysis of survival outcomes to treatment including pneumonitis as time-varying co-variate (**as primary analysis**) in the Cox proportional model. Please see below the figure illustrating the analysis. There was no difference on outcomes in univariable analysis. In addition, we also performed multivariable analysis (**Supplementary Table 3**).

We included the **Supplementary Table 3**, which include univariable and multivariable analysis and also the following statement:

Page 13 (lines 2-7) in the Results section:

“To account for lead-time bias, given the time-dependent nature of pneumonitis, we analyzed the impact of pneumonitis on survival outcomes by including pneumonitis as a time-varying co-variate in the Cox proportional hazard model and observed that the development of pneumonitis was not associated with mPFS (HR: 0.95 [95% CI, 0.63-1.43], P=0.79) or mOS (HR: 1.14 [95% CI, 0.70-1.87], P=0.60) (**Supplementary Table 3**).”

Supplementary Table 3. Univariable and multivariable Cox regression analysis.

Progression-free survival	Univariate Hazard Ratio [95%CI]	P-value	Multivariate Hazard ratio [95%CI]	P-value
Pneumonitis*				
(Discontinued vs. Not discontinued)	0.95 [0.63-1.43]	0.79	0.85 [0.51-1.42]	0.55
ECOG				
PS 0	-	-	-	-
PS 1	1.42 [1.03-1.95]	0.03	1.55 [1.03-2.34]	0.03
PS 2	1.76 [0.84-3.69]	0.13	3.37 [1.23-9.21]	0.02
Stage AJCC 8th				
IIIA	-	-	-	-
IIIB	1.62 [1.13-2.31]	0.008	1.48 [0.91-2.41]	0.11
IIIC	2.09 [1.34-3.27]	0.001	2.24 [1.26-4.00]	0.006
TMB Z-score**	0.66 [0.55-0.78]	<0.001	0.63 [0.53-0.76]	<0.001
PD-L1 TPS				
<1%	-	-	-	-
1-49%	1.16 [0.79-1.70]	0.45	1.21 [0.77-1.92]	0.40
50-89%	0.78 [0.50-1.21]	0.26	0.95 [0.59-1.58]	0.92
90-100%	0.36 [0.19-0.69]	0.002	0.43 [0.20-0.93]	0.03
NLR				
≥5	-	-	-	-
<5	0.63 [0.47-0.86]	0.004	0.87 [0.58-1.30]	0.49
Overall survival	Univariate Hazard Ratio [95%CI]	P-value	Multivariate Hazard ratio [95%CI]	P-value
Pneumonitis*				
(Discontinued vs. Not discontinued)	1.14 [0.70-1.87]	0.60	1.34 [0.69-2.60]	0.39
ECOG				
PS 0	-	-	-	-
PS 1	1.44 [0.91-2.26]	0.12	1.39 [0.76-2.54]	0.28
PS 2	2.85 [1.18-6.91]	0.02	3.67 [1.11-12.1]	0.04
Stage AJCC 8th				
IIIA	-	-	-	-
IIIB	1.34 [0.82-2.20]	0.24	1.46 [0.73-2.91]	0.28
IIIC	1.99 [1.09-3.66]	0.03	1.85 [0.82-4.16]	0.13
TMB Z-score**	0.80 [0.63-1.03]	0.08	0.82 [0.65-1.06]	0.14
PD-L1 TPS				
<1%	-	-	-	-
1-49%	0.80 [0.47-1.35]	0.40	0.61 [0.29-1.23]	0.16
50-89%	0.81 [0.46-1.43]	0.47	1.09 [0.54-2.21]	0.81
90-100%	0.31 [0.12-0.79]	0.01	0.40 [0.13-1.19]	0.09
NLR				
≥5	-	-	-	-
<5	0.61 [0.39-0.94]	0.02	0.76 [0.42-1.37]	0.36

*Pneumonitis: time-dependent adjusted
TMB Z-score* as continuous variable.

4. Discussion: “while prospective validation will further support our findings” – this may not occur. The authors indicated, “These data suggest that pneumonitis can result in insufficient durvalumab therapy which leads to poor survival. Strategies to minimize and reduce the risk of pneumonitis are critical as they can potentially improve patient outcomes”. The interpretation of early onset pneumonitis as causally related to poor outcomes is problematic as it may simply be a marker for the underlying mechanism that manifests in both early pneumonitis and poor prognosis.

We agree with the reviewer's comments regarding the above statements. We have edited and toned it down accordingly in the manuscript.

Page 19 (lines 10-14) in the Results section:

“Future studies, including either prospective validation studies or larger multi-institutional retrospective cohorts to build upon this work are necessary to further support our findings. Nonetheless, this is the largest report of clinicopathologic and genomic correlates of concurrent CRT and durvalumab efficacy in patients with stage III NSCLCs to date. ‘

Page 18 (lines 15-19) in the Results section:

”These data suggest that pneumonitis can result in insufficient durvalumab therapy which leads to poor survival; however, whether associated factors such as underlying lung disease may influence outcomes is unclear. Nonetheless, strategies to minimize and reduce the risk of pneumonitis are critical as they can potentially improve patient outcomes.”

REVIEWERS' COMMENTS

Reviewer #2 (Remarks to the Author):

I have no more comments

Reviewer #3 (Remarks to the Author):

The authors have addressed my concerns. Thank you.